# Fully Anonymous Decentralized Identity Supporting Threshold Traceability with Practical Blockchain

Submission Id: 2625

## ABSTRACT

Decentralized identity (DID) holds significant potential for applications in the Web3, such as digital markets and financial systems. Traditional DID paradigms offer a degree of privacy but struggle to prevent the link analysis on user behaviours and repeated public key usage. Anonymity is not fully achieved, as users' real identities or public keys are exposed to the issuing authority, while introducing high public key management complexity. Besides, existing anonymous credential schemes lack effective mechanisms for threshold traceability, not meeting the Web3's distributed governance requirements. In this paper, we propose FADID-TT, a **F**ully **A**nonymous **DID** system supporting **T**hreshold **T**racing with practical blockchain, to tackle the above challenges. Firstly, we propose a distributed identity registration scheme based on secret sharing. A committee composed of distributed issuing authorities is responsible for issuing user's secret key shares and no single entity in the system can obtain a user's real identity or public key, achieving anonymity to authority. Moreover, we design a *fully anonymous* DID system combined with anonymous signatures and decentralized anonymous credentials (DAC). A service provider can only use the committee public key to verify a user identity, eliminating the need for user public keys, fully resisting link attacks, and reducing the user public key management complexity from $O(n)$ to $O(1)$. Furthermore, we design a public verifiable *threshold tracing* mechanism that enables committee members to collaboratively trace the identity of a malicious user without compromising privacy guarantees. FADID-TT realizes publicly verifiable tracing via zero-knowledge proofs. Finally, we give comprehensive security analysis and concrete performance evaluation. In addition to evaluate each part of proposal, we also deploy FADID-TT on two well-known blockchain platforms including Hyperledger Fabric (permissioned) and Ethereum (permissionless) to demonstrate the practical feasibility of FADID-TT.

## CCS CONCEPTS

• **Security and privacy** → **Privacy-preserving protocols**; *Cryptography*; • **Information systems** → World Wide Web.

## KEYWORDS

decentralized identity, full anonymity, threshold traceability, blockchain

## 1 INTRODUCTION

Decentralized Identity (DID) is a new paradigm in digital identity management, allowing users to have greater control over their personal identity data and credentials through a decentralized infrastructure [1, 4]. DID improves security by reducing the risks of identity theft, privacy breaches, and data monopolization.

Owing to these remarkable properties, DID is regarded as a key element of the emerging Web3 ecosystem [18] and holds promise for a broad range of applications, including the secure digital asset ownership verification in the digital markets [12], the trust management in the financial system [7], the privacy-preserving personal data showcasing in the social medias [22], etc.

To support these diverse applications, the DID system relies on two key components: *Decentralized Identifiers* (DIDs) and *Credentials*. The DIDs, serving as unique identifiers for entities, are self-generated by users without relying on a central authority. The credential, consisting of verifiable claims, is a statement used to prove that an entity has some specific attributes, qualification, or identity [5]. These components work together to facilitate secure authentication and authorization, ensuring users are recognized and granted appropriate permissions in various applications.

As the application of DID is expanding, ensuring powerful privacy protection in DID systems becomes a critical concern. Notably, Fractal ID, a decentralized identity provider, experienced a significant breach of identity privacy data in July 2024, despite of its decentralized architecture[1]. This underscores the importance and urgency of protecting user privacy in DID systems.

*Anonymity* is one of the important privacy protection requirements in DID systems, which aims to prevent the disclosure of a user's real identity, thereby protecting users interests. Given stringent privacy regulations such as the GDPR [28], even a user's behavioral patterns are considered as personal data that needs to be protected. Thereby, we introduce the concept of *full anonymity* to meet high privacy standards. Specifically, full anonymity includes two key conditions: First, no party (including the verifier and issuing authority) can learn a user's real identity. Second, *unlinkability* must be guaranteed across multiple service interactions, preventing users from being profiled based on their behavior patterns. Besides, anonymous DID should support *traceability*, allowing authorities to reveal the true identity of users in cases of misconduct. For example, in the financial sector, anti-money laundering (AML) regulations require mechanisms to track suspicious activities while preserving user privacy. Additionally, the management of DID system is crucial, including the storage of user identities. However, existing DID solutions still face the following issues.

**Loss of full anonymity.** As shown in Figure 1, primary DID solutions like W3C DID [29] use an unique and fixed identifier to maintain the user's real identity from direct exposure. Nevertheless, they completely cannot resist the link analysis between different services via the same public key usage. Some existing researches, such as Pairwise DIDs [9] and CanDID [19], have already attempted to address this issue by generating distinct identifiers across different services, but the link analysis still occurs within a single application via the same public key usage, failing to achieve unlinkability. Although adopting the strawman one-time pseudonyms (i.e,

---

[1]https://www.biometricupdate.com/202407/data-breach-raises-questions-about-fractal-ids-decentralized-identity-architecture

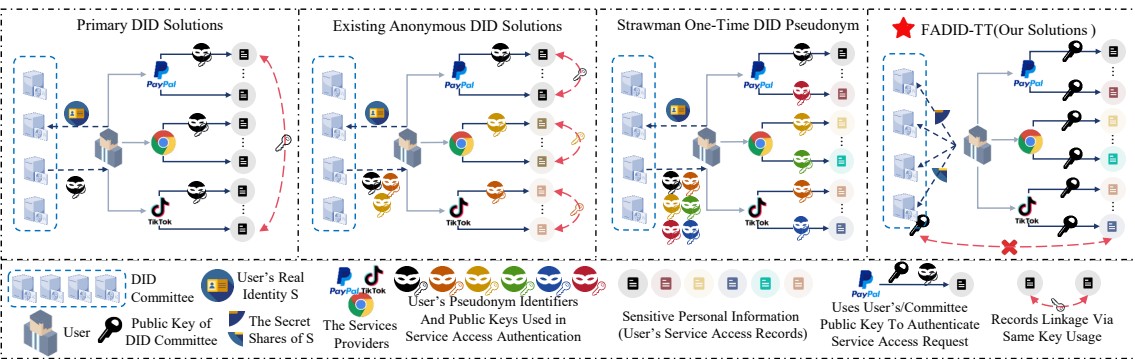

**Figure 1: Comparison between different DID schemes on full anonymity**

registering a new identifier for each service access) could achieve complete unlinkability, such method incurs significant overhead on DID committee. Each committee member needs to store and manage the whole mapping between the user real identity and pseudonyms, making it impractical for real-world deployment and losing anonymity to authorities.

**Lack of support for threshold traceability.** To maintain the user's real identity and attributes from direct exposure, decentralized anonymous credentials (DACs) [14] are proposed to achieve anonymity and unlinkability through cryptography mechanisms. DACs allow users to get verified without showing their real identities and attributes in an unlinkable way at the decentralized setting. However, most of existing DACs [27][24] ignore the accountability and traceability, failing to support the real-world governance requirements, let alone achieve threshold tracing in distributed Web3. In the limited work trying to achieve both privacy and traceability, TMAC [17] introduces a centralized tracing authority, making it vulnerable to the abuse of the tracing power and lack of transparency. While TABC [26] realizes threshold tracing, it exposes users' real identities during the registration process, which undermines anonymity to the authority and raises privacy concerns.

**High management complexity.** Additionally, in existing schemes, the registration authority needs to maintain a mapping between user real identity and their public key for authentication purposes. Consequently, it need store and manage at least $O(n)$ user public keys for $n$ users. This cost becomes considerable as the number of users in the system continues to increase, bringing high overhead.

Therefore, a burning question arise: *Can we achieve full anonymity in DID systems efficiently while enabling threshold tracing mechanisms for both privacy and accountability?*

**Our works.** To address the above issues, we propose **FADID-TT**, a **F**ully **A**nonymous **D**ecentralized **ID**entity system supporting **T**hreshold **T**racing with practical blockchains. Through introducing an anonymous DID committee of multiple issuing authorities called ADID committee, FADID-TT provides a distributed registration mechanism to offer stronger privacy for user's real-world identities and data. As shown in Figure 1, the user's real identity shares are distributed and managed among committee members via secret sharing, so as to realize anonymity of the real identity for all roles in the system, which means that even the ADID committee members less than threshold can not learn the real-world identity of the user (Table 1, Row 6, Column "Anonymity to authority").

Furthermore, we incorporate anonymous signatures and DACs to achieve the full anonymity in a decentralized setting. Upon registration, the user receives a secret key bound to their real identity and the committee's secret key, enabling anonymous signing. During the authentication process for service access, the service provider can only use the ADID committee's public key to verify the user identity even for different users, which eliminates the need for user public keys and fully resists link attacks (Table 1, "Anonymity to verifiers" & "Unlinkability"). Meanwhile, the cost of maintaining the mapping between the user's public key and the real identity is eliminated (Table 1, "User pk management complexity", $O(1)$).

Moreover, FADID-TT supports publicly verifiable threshold tracing, while retaining essential DAC properties such as selective disclosure, unlinkability, and anonymity, so that the identity and behavior patterns of the user can be protected. The main idea is introducing a distinct unlinkable tag during the presentation, which is derived from the one-time token and user's registration secret key, and can only be traced to the user's real identity through collaborative efforts of the ADID committee. By distributing the tracing power in ADID committee and setting a threshold, we prevent any single entity from compromising user privacy. Meanwhile, ADID committee could generate zero-knowledge proofs for tracing process, making it public verifiable and auditable (Table 1, "Threshold tracing"& "Verifiable traceability"). Furthermore, ADID committee can be deployed on practical blockchains [2, 30] to be compatible with existing distributed infrastructure.

**Contributions.** The contributions of our work are as follows.

- We propose a *distributed identity registration scheme based on secret sharing*, ensuring that no single entity in the system can obtain the user's real identity except the user himself. This registration only issues the user's registration secret key shares in a distributed manner and achieve anonymity to authority.

- We design a *fully anonymous DID system combined with anonymous signature and DAC*. Registered user's access can only be authenticated by the same committee public key, meanwhile the user attributes required for specific services can be hidden by DAC and each service access generates a one-time show token, completely resisting link analysis based on user's public keys or specific person data and achieving full anonymity. Simultaneously, we eliminate user's public key usage and decrease user's public key management overhead from $O(n)$ to $O(1)$.

- We design a *public verifiable threshold tracing mechanism*. Threshold tracing is supported by the distributed ADID committee without compromising the privacy properties of DAC, preventing

**Table 1: Comparison between FADID-TT and other related work**

| System | Full Anonymity | | | Traceability | | Management |
|---|---|---|---|---|---|---|
| | Anonymity to verifiers[1] | Anonymity to authority[2] | Unlinkability[4] | Threshold tracing [5] | Verifiable traceability | User pk management complexity[6] |
| CanDID[19] | ✔ | ✔ | ✘ | ✔ | ✔ | $O(mn)$ |
| Coconut[27] | ✔ | ✔[3] | ✔ | ✘ | ✘ | $O(n)$ |
| zk-creds[24] | ✔ | ✔[3] | ✔ | ✘ | ✘ | $O(n)$ |
| TMAC[17] | ✔ | ✘ | ✔ | ✔ | ✘ | $O(n)$ |
| TABC[26] | ✔ | ✘ | ✔ | ✔ | ✘ | $O(n)$ |
| **FADID-TT (Our work)** | ✔ | ✔ | ✔ | ✔ | ✔ | $O(1)$ |

[1] Anonymity to verifiers means verifiers can not learn the user real identity.
[2] Anonymity to authority means the user's real identity remains hidden to the authority even when user register with real identity considering regulatory.
[3] Coconut and zk-creds satisfy anonymity to authority when not considering tracing.
[4] Unlinkability means different accesses of the same user cannot be linked.
[5] In the column "Threshold tracing", ✔ represents centralized tracing, ✘ represents tracing is not supported.
[6] User pk management complexity means the number of user public keys that the authority needs to store, manage, and maintain, where $n$ is the number of users in the system, $m$ is the number of server providers, $k$ denotes the access count of a user per service.

the abuse of tracing power. Meanwhile, the tracing process is public verifiable via zero-knowledge proof, making it applicable to certain scenarios where accountability is important.

- We provide *security analysis, implementation on practical blockchains, and performance evaluation.* We give specific security definition and proof to illustrate our proposal satisfies the security goals. Concurrently, we conduct performance evaluation on the proposed schemes. Furthermore, we deploy FADID-TT on two well-known blockchain platforms including Hyperledger Fabric and Ethereum to demonstrate the practical feasibility.

## 2 PRELIMINARIES AND BUILDING BLOCKS

In this section, we briefly illustrate the foundational blocks necessary for FADID system construction including decentralized anonymous credentials, bilinear pairings, secret sharing, and Non-Interactive Zero-Knowledge Proof.

### 2.1 Decentralized Anonymous Credentials

Decentralized anonymous credential (DAC) schemes [14, 31] offer robustness and privacy for user-centered identity management in DID systems. DACs enable users to possess credentials issued by decentralized entities, thereby removing the need for online verification intermediaries. Moreover, DACs allow selective disclosure of personal attributes and empower users to anonymously authenticate themselves without revealing their actual identity or sensitive personal information. Meanwhile, the user can rerandomize its credentials independently to achieve unlinkability and ensure that the user cannot be traced. A general DAC scheme utilized in this work is formalized as the following algorithms:

- $DAC.Issue(req, isk) \rightarrow cred$: This algorithm handles the anonymous credential issuance process. Given a user's request $req$, decentralized issuers collaboratively sign the user's identity attributes using their issuing secret keys $isk$. It returns the anonymous credential $cred$ to the user.
- $DAC.Show(cred) \rightarrow stoken$ : This algorithm abstracts how anonymous credentials are used. The user (credential owner) inputs the issued credential $cred$, and outputs an one-time show token $stoken$. $stoken$ serves as a proof that the user owns a valid credential approved by the issuers without revealing actual identity information. $DAC.Show(cred)$ is non-deterministic which means the same credential can derive multiple different tokens. It ensures unlinkability across verification processes and allows the user to reuse credentials independently and maintain privacy.

- $DAC.Verify(stoken, ipk) \rightarrow 0/1$ : This verification algorithm checks the validity of the show token $stoken$ using the issuers' public key $ipk$. It returns 1 if the token is valid, and 0 otherwise.

### 2.2 Bilinear Pairing

Bilinear pairing is a kind of binary mapping on two cyclic groups. Let $\mathbb{G}$, $\widetilde{\mathbb{G}}$, and $\mathbb{G}_\mathbb{T}$ be three finite groups of prime order $p$. $g$ and $\widetilde{g}$ are the generators of $\mathbb{G}$ and $\widetilde{\mathbb{G}}$, respectively. A bilinear pairing is a map $e: \mathbb{G} \times \widetilde{\mathbb{G}} \rightarrow \mathbb{G}_\mathbb{T}$ with the following properties [20].

- Bilinearity: For all $g_1 \in \mathbb{G}$, $\widetilde{g_2} \in \widetilde{\mathbb{G}}$ and $u, v \in \mathbb{Z}_q^*$, $e(g_1^u, \widetilde{g_2}^v) = e(g_1, \widetilde{g_2})^{uv}$;
- Non-degeneracy: $e(g_1, \widetilde{g_2}) \neq 1$;
- Computability: There exists an efficient algorithm to compute $e(g_1, \widetilde{g_2})$ for all $g_1 \in \mathbb{G}$, $\widetilde{g_2} \in \widetilde{\mathbb{G}}$.

### 2.3 Secret Sharing

The $(n, t)$-threshold secret sharing technique allows a secret $s$ to be divided into $n$ parts called shares and the secret $s$ can only be reconstructed when at least correct $t$ shares are gathered. The Shamir secret sharing scheme (denoted as $SSS$) [25], which can achieve information-theoretical security is the most well-known and widely-used secret sharing method through Lagrange polynomial interpolation. The $(n, t) - SSS$ is comprised of two algorithms:

- $Share(s, n, t) \rightarrow \{s_i\}_{i=1}^n$ : The holder of secret $s$ selects $t - 1$ random coefficients $a_1, a_2, \ldots, a_{t-1}$ over a finite field $\mathbb{F}_p$ and construct polynomial $f(x) = s + \sum_{i=1}^{t-1} a_i x^i$. The secret $s$ satisfies $f(0) = s$. For each participant $i$, a corresponding share $s_i = f(i)$ is computed and securely distributed to the participant $i$.
- $Reconstruct(\{s_i\}_{i \in T}) \rightarrow s$ : Given any index subset $T$ of size greater than or equals to $t$, the secret $s$ can be reconstructed utilizing Lagrange interpolation. For each $i \in T$, the corresponding Lagrange coefficient $\lambda_i$ is computed as:

$$\lambda_i = \frac{\prod_{j \in T, j \neq i} j}{\prod_{j \in T, j \neq i} (j - i)}$$

Then through Lagrange interpolation, the secret $s$ can be reconstructed by computing $s = \sum_{i \in T} \lambda_i s_i$.

### 2.4 Non-Interactive Zero-Knowledge Proof

In a zero-knowledge proof (ZKP) system, there are two principal entities: the prover and the verifier. The prover's goal is to convince the verifier that a statement $R$ related to a secret value $w$ is

true, without revealing any additional knowledge about $w$. Non-Interactive Zero-Knowledge (NIZK) can achieve the following three fundamental properties [15] without interactive communication between the prover and verifier.

- *Completeness.* If the statement is true, an honest prover will be able to convince the verifier of its correctness.
- *Soundness.* If the statement is false, a dishonest prover cannot successfully convince the verifier of the statement's truth.
- *Zero-knowledge.* The interaction between the prover and the verifier reveals only that the statement is true, without disclosing any additional knowledge about the secret.

In the NIZK scheme, two core algorithms are involved: the prover generates a valid proof $\pi$ using the proving algorithm *Prove*, and the verifier checks the validity of $\pi$ with the corresponding verification algorithm *Verify* [13]. In the FADID-TT design, we adopt a non-interactive Sigma protocol [6]. We use the following notation to represent a NIZK proof $\pi$ generated for a secret value $w$ and a statement $R$ about $w$: $\pi \leftarrow NIZK_{SP}\{w : R\}$.

## 3 PROBLEM FORMULATION

In this section, we first outline the design goals to clearly demonstrate the problem our system is intended to solve or the specific outcomes we aim to achieve. Then, we introduce the main roles within the system and describe the system's workflow, showing how they interact together to reach the desired outcome. Afterward, we summarize the system's main functionalities through giving corresponding algorithm definition, providing a high-level understanding of the system's operational logic.

### 3.1 Design Goal

- **Decentralization.** The system must ensure decentralized trust and prevent single point of failure, thus enhancing fault tolerance and robustness. To achieve this, both the registration process and the tracing mechanism need operate in a distributed manner, ensuring that no centralized authority holds unilateral control, and the system remains resilient even if parts of it fail.
- **Full Anonymity.** The system aims to archive full anonymity which includes two key properties: anonymity to all parties within the system and unlinkability of different accesses to the service. The anonymity is to protect the user's real-world identity from being revealed to both verifiers and authorities within the system even during the registration. Meanwhile, the unlinkability is to ensure that a user's actions remain unlinkable across multiple interactions with the services so that users cannot be profiled based on analysis of their previous activities. Together, by achieving anonymity and unlinkability, the system can realize full anonymity to provide strong privacy.
- **Accountability.** While anonymity is important, the system also needs to support mechanisms for tracing and auditing under regulated conditions. If the user misuses the service, the system need be able to trace the user and reveal its real identity while ensure that user privacy is only compromised in compliance with legal or regulatory frameworks.
- **Security.** A primary focus of security in this system is to prevent identity forgery. The system must ensure that no entity can impersonate a user or forge credentials associated with the user's

real identity. This mainly includes the unforgeability of both identity secret key and credential.

### 3.2 System Model

**Roles.** There are four main actors in our system: users, ADID committee, verifiers (service providers) and credential issuers (authorities), as depicted in Figure 2.

- **Users.** Users are those individuals that want to access to services offered by online service providers. The users set in the system grows continuously. We consider that each user is associated with a unique identifier that already exists in the real world, such as an ID card number in CHN or a Social Security Number (SSN) in USA. For practical considerations, we assume that users may misbehave within the application after they obtain access to the service provided by the verifiers.
- **ADID Committee.** ADID Committee is a decentralized committee composed of distributed authorities which is responsible for user identity managment and satisfies the basic distributed threshold security assumptions. It is in charge of anonymous and decentralized registration of users, the audit accountability of users' real identities, and the threshold tracing of user's anonymous credentials. Given realistic privacy needs, it is assumed to be honest but curious.
- **Verifiers.** Verifiers are those entities that need to verify a user's identity information and credentials for authentication and authorization in Internet services, such as web service providers (Google, for example) and decentralized applications. We assume verifiers honest but curious, during the verification with the user.
- **Credential Issuers.** Credential issuers serve as decentralized authorities responsible for issuing credentials to users.

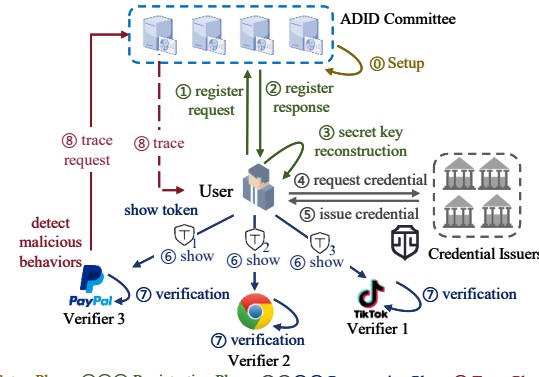

**Figure 2: Workflow of our FADID-TT system**

**Work Flow.** Figure 2 shows the basic workflow of the system, where the credentials component we instantiate by existing anonymous credential system. It can be divided into four main phases: setup, registration, presentation, and tracing.

*Setup.* Initially, setting global cryptography parameters essential for operation in the system, the ADID committee distributedly generate it implicit secret key and public verification key without relying on a trusted third party (Figure 2, step ⓪).

*Registration.* Given the situation that a new user Alice wants to access a verifier's (e.g. YouTube) services, she first anonymously sends an identity registration request to the ADID committee, which

is bound to her unique identifier that already exists in the real world (Figure 2, step ①). Then, the ADID committee verifies the user's identity information and the uniqueness of the registration through local data table. After confirming that the information is both legitimate and unique, the ADID committee distributedly computes the share of user's registration secret key, which is associated with the user's real identity and the committee's secret key. This key share is subsequently distributed to the user (Figure 2, step ②). Next, upon collecting enough key shares that meet the reconstruction conditions, the user reconstructs the complete registration secret key locally and uses ADID committee's verification key to validate its legitimacy (Figure 2, step ③). After successful verification, the registration process is completed, and it is worth noting that the user only obtains the user's private key after registration, but not the user's public key.

*Presentation.* Afterward, the user can apply for an anonymous credential from the credential issuers (Figure 2, step ④). Once the credential issuers verifies the legitimacy of the user's identity attributes, it will issue the anonymous credential to the user in a distributed manner (Figure 2, step ⑤). After that, the user randomizes the received credential locally to generate a one-time show token, which is to proves to the verifier (service provider) that her identity attributes have been certified by the credential issuers (authorities), in a unlikable way between different verification process. Meanwhile, the user signs the show token using the previously registered secret key and sends it along with the signature to verifier (Figure 2, step ⑥). And then, the verifier verifies the show token without contacting the credential issuers directly and utilizes ADID committee's public verification key to verify the legitimacy of the signature (Figure 2, step ⑦). Upon successful verification, the legitimacy of the user's identity can be confirmed, allowing the user to receive the corresponding services from the verifier.

*Tracing.* Besides, considering the potential for malicious use of anonymous credentials or post-audit scenarios to trace the real identity of credential holders, the ADID committee, based on credential tracing requests that may be proposed by verifier, can conduct identity threshold tracing to reveal the real identity of the holder and generate a proof so that this tracing process can be publicly verified. And then, this tracing record will be publicly uploaded on-chain to warn against malicious behavior (Figure 2, step ⑧).

## 4 FADID DESIGN

In this section, we detail our design of the system FADID-TT.

### 4.1 Concrete Construction

Our scheme operates through four phases: setup, registration, presentation, and tracing. We omit the processes of anonymous credential issuance and obtaining, as these are not the primary focus of our work, which has been extensively covered in existing literature [11, 14, 21, 24, 27]. Moreover, our FADID system is designed to flexibly integrate with various multi-show anonymous credentials systems. Each of the four phases comprises several algorithms, which we detail below. Figure 7 in appendix illustrates the interactions among parties utilizing these algorithms within our system.

#### 4.1.1 Setup Phase.

To set up the system initially, first the **GlobalSetup** function determines global cryptographic parameters essential for operation in the system, then the ADID committee employs the **CKGen** function to distributedly generate key without relying on a trusted third party. The algorithms are implemented as follows.

- **GlobalSetup**$(1^\lambda) \to pp$. Given the input of a security parameter $\lambda$, the algorithm generates a bilinear pairing description $bp = (p, \mathbb{G}, \widetilde{\mathbb{G}}, \mathbb{G}_T, e, g, \widetilde{g})$, where $\mathbb{G}, \widetilde{\mathbb{G}}, \mathbb{G}_T$ are finite groups with large prime order $p$. The mapping $e : \mathbb{G} \times \widetilde{\mathbb{G}} \to \mathbb{G}_T$ represents a type-3 bilinear mapping, with $g$ and $\widetilde{g}$ as the generators of group $\mathbb{G}$ and $\widetilde{\mathbb{G}}$ respectively. In addition, it sets up the threshold parameters for the ADID committee with $(n, t)$, where $n$ is the number nodes in ADID committee and $t$ denotes the threshold for ADID committee participants. The output global public parameters are given by $pp = (bp, (n, t))$.

- **CKGen**$(pp) \to (\{csk_i\}_{i=1}^n, cvk)$. Taking the global public parameters $pp$ as input, the ADID committee runs a the distributed key generation (DKG) protocol [16]. Through the DKG protocol, all $n$ nodes of the ADID committee collectively generate a group public key $cvk$ (with an implicit group secret key $csk$ which is never computed explicitly) and satifies the relation $cvk = g^{csk}$. Also, each node $i$ within the ADID committee gets its individual secret key $csk_i$, which constitutes a share of $csk$ in $(n, t)$ Shamir secret sharing scheme.

#### 4.1.2 Registation Phase.

To register in the system, the user invokes **PrepareReg** function using her unique identifier $s$ which may be an identity card number, a SSN, or a Decentralized Identifier (DID) in other DID systems. Subsequently, this registration request is distributed to the ADID committee by the user. After that, the ADID committee executes **UKeyIssue** algorithm to first verify the legitimacy and uniqueness of the request and then collaboratively issue shares of the user's secret key, denoted as $ursk_i$. Upon receiving a sufficient set of registration secret key shares that fulfill reconstruction criteria, the user employs **UKeyRecon** to derive the user's complete registered secret key $ursk$. Finally the user runs **UKeyVerify** locally to confirm the correctness of the reconstructed secret key. Detailed implementations of these algorithms are provided below.

- **PrepareReg**$(s) \to regReq$. The user inputs her unique identifier $s$, which already exists in the real world (e.g. a government-issued ID number). Then it runs $\{s_i\}_{i=1}^n \leftarrow Share(s, n, t)$ to distribute the user's real-world identity $s$ among the ADID committee nodes using a Shamir threshold secret sharing scheme. Meanwhie, the user makes a Feldman commitment to her indentity in the form of $g^s$, which is sent to the ADID committee. So after that, each node within the ADID committee receives a share of the user's identity and the corresponding commitment. Crucially, this ensures that the ADID committee, even if partially compromised, cannot deduce the user's real identity. The algorithm outputs the registration request information $regReq = (\{s_i\}_{i=1}^n, g^s)$ sent to the ADID committee.

- **UKeyIssue**$(\{csk_i\}_{i=1}^n, regReq) \to \{ursk_i\}_{i=1}^n$. The ADID committee parses the registration request information $regReq$ of a new user as $(\{s_i\}_{i=1}^n, g^s)$. The identity commitment $g^s$, included in the registration request, is checked against the locally recorded data table $\mathcal{T}_i$ on each node $i$ of the ADID committee. This table

$\mathcal{T}_i$ of node $i$ stores the registration information for all legal users who had been registered on the ADID committee. Initially these tables an empty, i.e $\mathcal{T}_i = \varnothing$ for $i = 1, \ldots, n$. If the identity commitment $g^s$ is not found in any local data table across all ADID committee nodes, i.e., $g^s \notin \mathcal{T}_i$ for $i = 1, \ldots, n$, the uniqueness of the new user's identity is confirmed. Otherwise, it aborts.

Afterward, the ADID committee proceeds to issue a secret key to be registered, $ursk = (g^{\frac{1}{s+csk}}, \widetilde{g}^{\frac{1}{s+csk}})$, to the user in a distributed manner. This user secret key is derived in form of a distributed verifiable random function's output (DY-VRF) [10], bound to both the user's real identity $s$ and the ADID committee's secret key $csk$. The process leverages the linear homomorphism of Shamir secret sharing and the multiplication protocol for Shamir secret sharing [3].

Specifically, the distributed computation of the user secret key proceeds as follows: (1) Each node $i$ of the ADID committee computes $\mu_i = csk_i + s_i$ using the user identity share $s_i$ and its secret key $csk_i$. (2) A random number $\rho$ is implicitly generated among the $n$ nodes of the ADID committee in a distributed manner, that is, each node $i$ only gets its corresponding share $\rho_i$. (3) Runing the Shamir secret sharing multiplication protocol [3], each node $i$ gets the share of the product of $\rho$ and $\mu$, represented as $(\mu\rho)_i$, using $\mu_i$ and $\rho_i$ without revealing the full value of $\mu$ or $\rho$. (4) The ADID committee collectively recovers $\mu\rho$ using Lagrange interpolation: $\mu\rho = \sum_{i=1}^{t}(\lambda_i(\mu\rho)_i)$, where $\lambda_i$ is the Lagrange coefficient. And the value $\mu\rho$ is exposed publicly among the ADID committee. (5) Each node $i$ computes locally its share of the inverse of $\mu$, denoted as $(\frac{1}{\mu})_i = \frac{\rho_i}{\mu\rho}$. (6) Each node $i$ computes its corresponding share of user secret key: $ursk_i = (usk_i, \widetilde{usk}_i)$, where $usk_i = g^{(\frac{1}{\mu})_i}, \widetilde{usk}_i = \widetilde{g}^{(\frac{1}{\mu})_i}$.

Finally, each node $i$ of the ADID committee respectively sends its share of the user's registered secret key $ursk_i$ to the user, ensuring the secure and distributed generation of the registration secret key. And each node $i$ updates its recorded registration data table $\mathcal{T}_i$ off chain by adding the registered and verified request $regReq$ to $\mathcal{T}_i$.

- **UKeyRecon**($\{ursk_i\}_{i=1}^{t}$) → $ursk$. The user parses the received secret key share $ursk_i$ from node $i$ as $(usk_i, \widetilde{usk}_i)$. Upon collecting a sufficient set of registration secret key shares of size up to the threshold $t$, the user aggregates them into user's secret key $ursk = (usk, \widetilde{usk}) = (\prod_{i=1}^{t}(usk_i)^{\lambda_i}, \prod_{i=1}^{t}(\widetilde{usk}_i)^{\lambda_i})$ This aggregation leverages Lagrange interpolation, and $\lambda_i$ represents the Lagrange coefficient.
- **UKeyVerify**($cvk, ursk$) → $0/1$. For the reconstructed user's secret key $ursk$ parsed as $(usk, \widetilde{usk})$, the user verifies its validity using the ADID committee's verification key $cvk$. The verification is performed by checking the following equations:

$$e(cvk \cdot g^s, \widetilde{usk}) = e(g, \widetilde{g}) \tag{1a}$$

$$e(usk, \widetilde{g}) = e(g, \widetilde{usk}) \tag{1b}$$

Equation (1a) verifies the correctness of $\widetilde{usk}$, equation (1b) verifies the consistency between $\widetilde{usk}$ and $usk$. Together, these conditions validate the reconstructed secret key $ursk$. It returns 1 if both equations hold, indicating $ursk$ is valid; otherwise, it outputs 0.

### 4.1.3 Presentation Phase.

After successful registration, the user gets her registered secret key $ursk$. Notably, there is no associated user public key; instead, our system utilizes the committee's verification key $cvk$. As outlined earlier, we assume the deployment of a Decentralized Anonymous Credential (DAC) system, where the user has already obtained a valid anonymous credential $cred$, issued by credential issuers through the function $DAC.Issue(req, isk) \rightarrow cred$. Hence, during the presentation phase, the user invokes **TokenPresent** algorithm to show the $cred$ for the verifier. Then the verifier runs **VerPresent** to verify this presentation.

- **TokenPresent**($ursk, cred$) → $(stoken, T_s, \pi_{T_s})$. The user calls $DAC.Show(cred) \rightarrow stoken$ function to generate a one-time show token $stoken$ of her owned credential $cred$. Then, the user signs $stoken$ using the previously registered secret key $ursk$. To be specific, parsing $ursk$ into $(usk, \widetilde{usk})$, the user computes the tag $T_s = e(stoken, \widetilde{usk})$, which operates as a verifiable unpredictable function. After that, the user constructs the proof $\pi_{T_s}$ to demonstrate the well-formedness of $T_s$ without leaking the secret key $ursk$, using the method similar to the anonymous signature scheme SyRA [8]. It is noticeable that, for a user, although the output tag $T_s$ is deterministically derived from $stoken$, yet the onre-time show token $stoken$ is pseudorandom and appears different across multiple presentations. Thus this still guarantees the unlinkability of multiple uses of the same credential, preventing any correlation between them. Besides, this design contrasts with SyRA [8], where the tag takes the form $T = e(cxt, \widetilde{usk})$, making different presentations of the credential under the same context $cxt$ linkable.
- **VerPresent**($cvk, stoken, T_s, \pi_{T_s}$) → $0/1$. The verifier employs the algorithm $DAC.Verify(stoken, ipk) \rightarrow 0/1$ to verify the received show token $stoken$. If the token is valid (the result is 1), then the verifier proceeds to verify the corresponding tag $T_s$ and proof $\pi_{T_s}$ the assistance of the ADID committee's verification key $csk$, using the method in SyRA [8]. Only if both verifications succeed it returns 1; 0 otherwise.

### 4.1.4 Tracing Phase.

Upon the presentation is successfully verified, the authenticity of the user's identity is confirmed. Thereby the user is granted access to the services provided by the verifier. However, certain scenarios necessitate credential tracing. For instance, when anonymous credentials are misused, or post-audit investigations require revealing the credential owner's real identity, it becomes crucial to identify users who exploit the service or engage in misconduct after gaining access. In such cases, the service provider must be able to identify the credential holder. Therefore, to address this, the credential tracing requestor (such as the verifier in the example above or other relevant parties) invokes **PrepareTrace** function to generate an anonymous credential tracing request. This request is submitted to the ADID committee for further action. Upon receiving the request, the ADID committee executes the **Trace** algorithm to validates it and then threshold-open the real identity of the credential holder. At the same time, a publicly verifiable proof $\pi_{trace}$ is generated. Any participant in the system can subsequently employs **TraceVer** to verify the correctness of the tracing process.

- **PrepareTrace**$(stoken, T_s, \pi_{T_s}) \rightarrow complainReq$. The verifier, when detecting malicious behavior exhibited by the user after service access is granted, can initiate a tracing process. Using the corresponding show token $stoken$, tag $T_s$ and proof $\pi_{T_s}$ from the previous presentation, the verifier creates a complaint request, $complainReq$, which includes a description of the requestor's identifying information and potential evidence of credential misuse. Then, the verifier sends the package $complainReq, stoken, T_s, \pi_{T_s}$ to the ADID committee for tracing.

- **Trace**$(\{csk_i\}_{i=1}^t, complainReq, stoken, T_s) \rightarrow (s, \pi_{trace})$. The ADID committee first checks the received complaint request $complainReq$. After conforming its legitimacy, $t$ nodes from ADID committee collaborate using their secret keys $\{csk_i\}_{i=1}^t$ to trace the credential holder associated with the show token $stoken$ and tag $T_s$. The specific tracing process is as follows: (1) Every node within the ADID committee computes $T_{tk} = e(stoken, \widetilde{g})$. (2) Each node $i$ of the ADID committee retrieves in its registration data table $\mathcal{T}_i$ one by one for an entry that satisfies specific conditions. For better description of the algorithm process, we assume the table is indexed. When checking the entry $\langle k \rangle$ in the registration data table, it does: (i) Each node $i$ of the ADID committee computes $T_{tk,i}^{\langle k \rangle} = T_s^{s_i^{\langle k \rangle} + csk_i}$ and broadcasts $T_{tk,i}^{\langle k \rangle}$ to the committee. (ii) Using the Lagrange interpolation where $\lambda_j$ represents the Lagrange coefficient, each node $i$ computes $T^{\langle k \rangle} = \prod_{j=1}^t (T_{tk,j})^{\lambda_j}$. (iii) Each node $i$ checks whether the equation $T^{\langle k \rangle} = T_{tk}$ holds or not. If the equality is true for all the $t$ nodes, the searching loop ends. If not, ADID nodes continue to check the next entry. We denote the entry which satisfies above condition as $\langle * \rangle$. By the way, if no such entry exists, the algorithm aborts. Once identified, the ADID committee executes $Reconstruct(\{s_i^{\langle * \rangle}\}_{i \in [t]}) \rightarrow s^{\langle * \rangle}$ to reconstruct the real indentity of the credential holder. And each node $i$ computes proof $\pi_{T_{tk,i}} = NIZK_{SP}\{(s_i^{\langle * \rangle}, csk_i) : T_{tk,i}^{\langle * \rangle} = T_s^{s_i^{\langle * \rangle} + csk_i}\}$, which is to ensure the well-formedness of $T_{tk,i}^{\langle * \rangle}$ in the tracing process, preventing malicious adversary fabricating intermediate evidence. The result is $s = s^*$ and $\pi_{trace} = (\{T_{tk,i}^{\langle * \rangle}\}_{i=1}^t, T_{tk}, \{\pi_{T_{tk,i}}\}_{i=1}^t)$.

- **TraceVer**$(cvk, \pi_{trace}) \rightarrow 0/1$. Parsing the tracing proof $\pi_{trace}$ as $(\{T_{tk,i}^{\langle * \rangle}\}_{i=1}^t, T_{tk}, \{\pi_{T_{tk,i}}\}_{i=1}^t)$, it verifies the $\pi_{T_{tk,i}}$ using the verification algorithm of sigma protocol for $i = 1, \ldots, t$. The algorithm outputs 1 if the proof is valid, 0 otherwise.

## 4.2 Security Analysis

### 4.2.1 Security Definition.

We first give a definition of security properties in FADID-TT.

**Definition 1** (*Full Anonymity.*) A DID system is said to provide full anonymity if and only if the following properties are satisfied:

- *Anonymity to authorities and verifiers.* Users remain completely anonymous with respect to both registration authorities and verifiers. Registration authorities cannot learn the user's true identity during registration, and verifiers are unable to extract any identifying information during verification. Thus, the user's identity is hidden from all system actors.

- *Unlinkability.* Users can engage with multiple verifiers without those verifiers being able to link the interactions to the same identity. This prevents different verification interactions, even when involving the same verifier, from being traced back to a single user or credential.

**Definition 2** (*Traceability.*) Cooperation among threshold honest ADID nodes can always correctly trace the owner of the credentials through a valid credential show token, by executing the tracing algorithm.

**Definition 3** (*Public Verifiability of Tracing.*) Without revealing sensitive information or violating privacy, any party can verify whether a specific tracing process has been correctly executed. This ensures that an honest user cannot be wrongly accused in the tracing process.

**Definition 4** (*Unforgeability.*) This includes: (1) Unforgeability of registration: An adversary cannot forge a valid registration secret key for a user without knowledge of the ADID committee's secret key. (2) Unforgeability of presentation: Users who have not a valid user secret key $ursk$ issued by by the ADID committee in registration phase, along with a valid credential from the credential issuers, cannot forge a valid presentation that pass verification.

**Definition 5** (*Correctness.*) Our scheme is correct if (1) the user's registration secret key $ursk$ should be accepted provided that $ursk$ is reconstructed from the shares issued by $t$ honest nodes within ADID committee; (2) a presentation generated by legitimate users using the TokenPresent algorithm should be accepted by the VerPresent algorithm, provided the $cred$ is valid; (3) by running the tracing algorithm, any $t$ honest tracers in committee should output the same user's real identity $s$ and a valid proof $\pi_{trace}$ that can be accepted by the TraceVer algorithm.

### 4.2.2 Security Proof.

We now provide a brief security proof of FADID-TT and illustrate how it satisfies the security properties defined above.

- **Full Anonymity.** The anonymity to registration authorities is guaranteed by the perfect secrecy of the Shamir's secret sharing. And the anonymity to verifiers follows from the anonymity of DACs and anonymous signature that do not require the user's public key. In addition, the unlinkability is ensured by the randomization of credential during the presentation phase and the pseudo-randomness of the associated tag.

- **Traceability.** The one-time token owner (malicious user) inevitably meets the matching criteria in tracing process, and the registration phase ensures each user in the system has unique corresponding real identity through duplication, so the tracing algorithm can accurately trace the one-time token owner.

- **Public Verifiability of Tracing.** This property is supported by the zero-knowledge property and soundness of non-interactive zero-knowledge proofs, allowing verification of the correctness of a tracing process without compromising privacy.

- **Unforgeability.** This follows from the unforgeability properties of the verifiable random function, DACs, and the signature.

- **Correctness.** This is satisfied by the correctness of each algorithm of our design. Roughly speaking, every output generated by an honest execution according to the design specifications will be accepted by the corresponding verification algorithm.

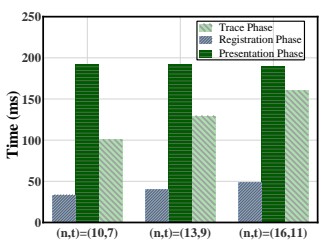

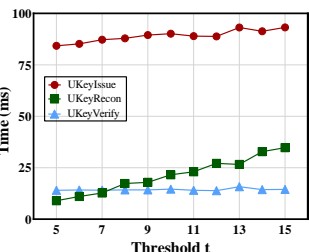

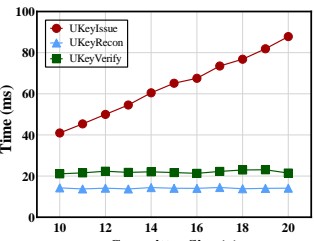

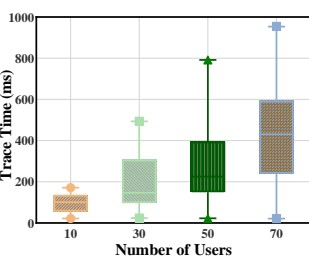

Figure 3: Runtime of different phases of the system.

Figure 4: Runtime of algorithms with different $t$.

Figure 5: Runtime of algorithms with different $n$.

Figure 6: Tracing time for different number of users.

## 5 IMPLEMENTATION AND EXPERIMENTS

**Experimental Settings.** All experiments were conducted on a virtual machine on a personal computer, which operates with Ubuntu 22.04.4 LTS as its operating system, featuring an 11th Gen Intel(R) Core(TM) i7-11800H CPU @ 2.30GHz and 4GB of RAM.

**Performance Evaluation.** We first implement the functionalities described in section 4.1 in Golang using the PBC[2] library, where the bilinear pairing is the D-type pairing defined in PBC. For the DAC component, we instantiate it with a threshold issuance version of the anonymous credential scheme proposed in [23], where the user credential is fixed to 10 attributes. And we test the performance at approximately the ECC-200-bit security level. We conduct performance evaluation on each part of our proposal.

Figure 3 shows the runtime of the main phase in our system, under different size of ADID committee with fixed fault tolerance ratio of $n = 3f + 1, t = 2f + 1$ ($f$ is fault node number). Setup phase is excepted due to it is only executed once at the initialization of the system. The result in Figure 3 illustrates that: (1) Although the time for registration and tracing phase increases with the growth of ADID committee, the running time of the main phases in our system is within the millisecond range and acceptable for common personal devices. (2) The execution time of the presentation phase remains stable with varying of the ADID committee size, because the verifier only needs to use the fixed ADID committee public key to authenticate the user who accesses services with the credential.

Figure 4 and Figure 5 depict the relationship between the key algorithms in the registration phase and two parameters including the number of ADID committee nodes $n$ and the threshold $t$. In Figure 4 $n$ is fixed at 20, and in Figure 5 $t$ is fixed at 10. From these figures, we can make the following observation : (1) The **UKeyIssue** computation time exhibits a linear increase with the $n$, but remains stable by $t$. (2) The **UKeyRecon** computation time is independent of $n$ but increases linearly with $t$. (3) The **UkeyVerify** computation time is invariant with respect to both $n$ and $t$. The reason is that all the committee nodes participate in the distributed user registration key issuance process **UKeyIssue**, requiring $n$ individual computation of key shares. However, users only need to gather $t$ key shares to reconstruct the full secret key in **UKeyRecon**. Meanwhile, the user only need verify one reconstructed key in **UkeyVerify** which has no relation to both $n$ and $t$.

Figure 6 illustrates the time consumption for tracing a random malicious user under different total numbers of registered users in the system, with the ADID committee's threshold set to $(n, t) = (10, 7)$. The results demonstrate that although the tracing time

fluctuates for individual cases, the tracing time increases as the growth of registered users in terms of the overall trend. This can be attributed to the fact that the search space expands with the scale of the system. But the longest tracing time is less than 1 second.

**Deployment on Blockchain.** We deploy FADID-TT on two well-known blockchain platforms including Hyperledger Fabric (permissioned setting) and Ethereum (permissionless setting) to demonstrate the practical feasibility with existing distributed infrastructure. In deployment on Hyperledger Fabric, we deploy ADID committee nodes as Fabric peers and establish a consortium chain. In deployment on Ethereum, we implement system operations as smart contracts. The ADID committee's threshold in both deployment is set to $(n, t) = (10, 7)$.

As Table 2 shows, in registration and presentation phase test, the ADID committee members work off-chain, so the result is similar to Figure 3. The main on-chain cost happens in the setup phase (because the ADID committee public verification key $cvk$ needs to be recorded on the blockchain) and the trace phase (because the ADID committee records the real identity of malicious users and public verifiable proof on the blockchain). For Hyperledger Fabric, the time cost of on-chain operations is approximately 2 seconds. On Ethereum, the gas consumption of on-chain operations is around 90,000 (about $3.5). The results illustrates that our FADID-TT system is highly practical.

**Table 2: FADID-TT time/gas cost on the Fabric/Ethereum**

| Operation | Setup | Registration | Presentation | Trace |
|---|---|---|---|---|
| **Fabric** (time) | 2.29362s | 40ms (Off-chain) | 190ms (Off-chain) | 2.06799s |
| **Ethereum** (gas) | 92470 | | | 92652 |

## 6 CONCLUSION

In DID, achieving powerful privacy protection practically while enabling accountability is challenging. We present FADID-TT, a fully anonymous decentralized identity system supporting threshold tracing with practical blockchain implementations. By utilizing DACs and anonymous signatures, FADID-TT achieves full anonymity to offer strong privacy protection for users while maintaining accountability through a publicly verifiable threshold tracing mechanism. Besides, our solution effectively reduces the complexity of user public key management. Experiment indicates the practical feasibility of our system. Future work can further build on these insights.

[2]https://github.com/Nik-U/pbc

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

## A  FUNCTIONALITIES

In order to illustrate the system functionalities more clearly, we present the formal definition of the main algorithms to realize these functionalities below:

- **GlobalSetup**$(1^\lambda) \rightarrow pp$. The algorithm takes as input the security parameter $\lambda$, and outputs the global public parameters $pp$. The public parameters $pp$ are implicit inputs to all other algorithms.
- **CKGen**$(pp) \rightarrow (\{csk_i\}_{i=1}^n, cvk)$. This algorithm is executed by the ADID committee, where $n$ is the number of ADID committee nodes and $t$ is the threshold of ADID committee. Taking public parameters $pp$ as input, it outputs secret key $csk_i$ for each committee node, and a joint verification key $cvk$ among the ADID committee.
- **PrepareReg**$(s) \rightarrow regReq$. The user takes as input her unique identifier $s$, which already exists in the real world. The algorithm outputs the registration request information $regReq$ sent to the ADID committee.
- **UKeyIssue**$(\{csk_i\}_{i=1}^n, regReq) \rightarrow \{ursk_i\}_{i=1}^n$. Each node of the ADID committee utilizes its secret key $csk_i$ and the registration request $regReq$ to generate the share of the user's secret key $ursk_i$.
- **UKeyRecon**$(\{ursk_i\}_{i=1}^t) \rightarrow ursk$. The reconstruction operation is performed by the user, which takes $t$ shares of user's registration secret key as input and aggregates them into user's registered secret key $ursk$.

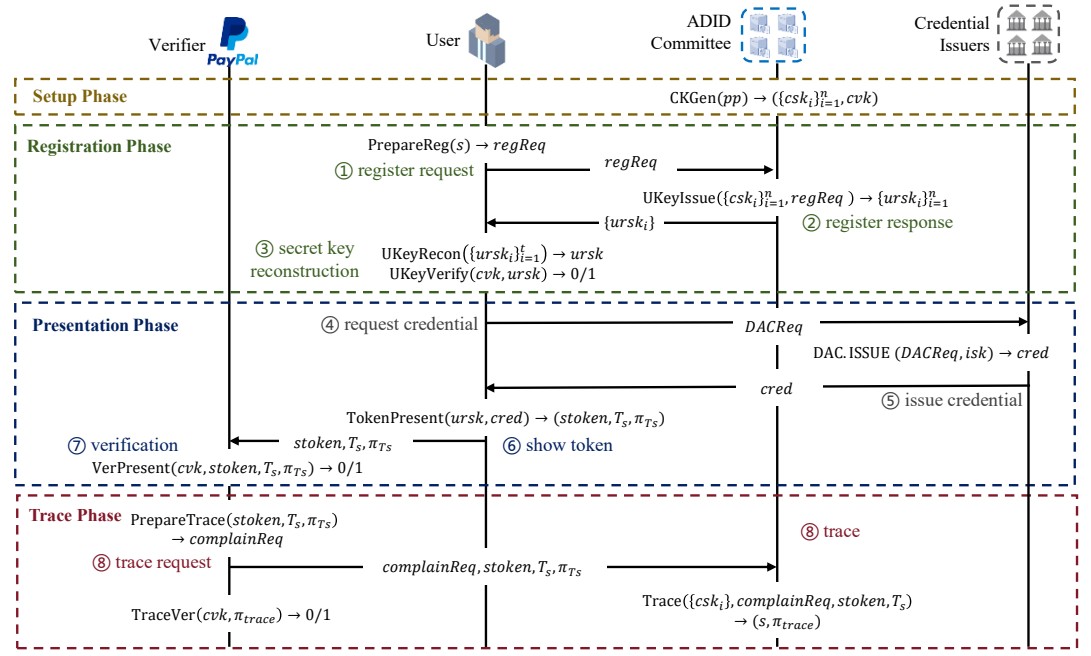

**Figure 7: Concrete phases and steps of FADID-TT**

- **UKeyVerify**($cvk, ursk$) → $0/1$. The algorithm is run by the user, which takes as input the committee's verification key $cvk$ and the reconstructed user's registered secret key $ursk$. It outputs 1 if $ursk$ is valid, 0 otherwise.
- **TokenPresent**($ursk, cred$) → ($stoken, T_s, \pi_{T_s}$). The user inputs her registered secret key $ursk$ and the credential $cred$ obtained from the credential issuers. The output consists of the show token $stoken$ of the credential and an anonymous signature of the show token which includes a tag $T_s$ and its corresponding proof $\pi_{T_s}$.
- **VerPresent**($cvk, stoken, T_s, \pi_{T_s}$) → $0/1$. The algorithm is run by the verifier, which takes as input the committee's verification key $cvk$, the received show token $stoken$, the corresponding tag $T_s$ and proof $\pi_{T_s}$. It output 1 if the presentation is verified, 0 otherwise.
- **PrepareTrace**($stoken, T_s, \pi_{T_s}$) → $complainReq$. The algorithm is executed when the verifier detects malicious behavior exhibited by the user after the show token is validated and the service has been rendered. The verifier creates a complaint request $complainReq$ to trace the credential holder responsible for the malicious actions, using the corresponding show token $stoken$, tag $T_s$ and proof $\pi_{T_s}$ from the previous presentation.
- **Trace**($\{csk_i\}_{i=1}^{t}, complainReq, stoken, T_s$) → ($s, \pi_{trace}$). The algorithm is run by the ADID committee. It verifies the complaint request $complainReq$. Then $t$ nodes from ADID committee collaborate using their secret keys $\{csk_i\}_{i=1}^{t}$ to trace the credential holder associated with the show token $stoken$ and tag $T_s$. It returns the corresponding user's real identity $s$ along with the proof $\pi_{trace}$ that verifies the correctness of the tracing process.
- **TraceVer**($cvk, \pi_{trace}$) → $0/1$. This algorithm can be executed by any party. It takes as public input the ADID committee's

verification key $cvk$ and the tracing proof $\pi_{trace}$, returning 1 if the proof is valid, 0 otherwise.

## B USE CASE

We will give a use case on FADID-TT system in decentralized finance application (Defi) of Web3. For a financier in Defi, he should prove to an investor that he has a certain identity and that the identity has corresponding asset collateral. But at the same time, the financier would be unwilling to disclose his private information including specific collateral amount and identity to investor. Furthermore, when the investor discovers malicious or unpurposed behavior by the financier, such as money laundering or market manipulation, financier identity can be found out by the investor.

As shown in Figure 7, an on-chain ADID committee has executed setup operations to generate system private key and public key. After that, the ADID committee can provide the identity registration functions. Besides, we assume the trusted distributed credential issuers can issue DAC for the financier. The financier (performing as user in FADID-TT) and the investor (performing as verifier in FADID-TT) can trust the ADID committee and DAC issuers. The financier first executes registration operation and interacts with ADID committee to get $ursk$. Then the financier requires credentials to prove his collateral amount by ZKP from credential issuers. After that, the financier rerandomizes the credential to generate $stoken$. Through verifying $stoken$, the investor can make sure that the financier is registered by ADID committee and has enough collateral. When the investor discovers malicious behavior by financier, it can send trace request to ADID committee and the tracing result can be recorded on the blockchain as further evidence of accountability.

