# OpenReview forum: "Fully Anonymous Decentralized Identity Supporting Threshold Traceability with Practical Blockchain"
_ACM.org/TheWebConf/2025/Conference — WWW 2025 Poster_

### Official Review · Reviewer_hrfq · 2024-11-28

**Novelty:** 5
**Technical Quality:** 5

**Review:**

This paper proposes a Fully Anonymous Decentralized Identity Supporting Threshold Traceability, which effectively addresses the issues of full DID and accountability in decentralized Web3.

Pros:

1. Compared to existing works, this scheme leverages anonymous signatures and DAC to achieve full anonymity for verifiers and authorities with high efficiency. In addition, the scheme provides a threshold tracing mechanism that supports privacy protection and resists identity forgery.
2. The writing of the paper is excellent, with clear expression, well-structured methodology, as well as rigorous analysis and experiments.
3. The implementation of FADID-TT on well-known blockchain platforms (including Hyperledger Fabric and Ethereum) and the use case on the decentralized finance application (Defi) of Web3, demonstrate the practical feasibility of  FADID-TT.

Cons:

1. The font size in the figures in the paper is slightly small, which affects the reading experience. It is recommended to modify the presentation of the figures.
2. The authors emphasize that their work is efficient. Therefore, the experimental section should include comparisons of performance, such as storage and computational overheads at different stages, with existing related works.
3. Is the comparison with related works sufficiently comprehensive? At least, we find that papers ([DCAP: A Secure and Efficient Decentralized Conditional Anonymous Payment System Based on Blockchain] and [AttriChain: Decentralized traceable anonymous identities in privacy-preserving permissioned blockchain]) are related works, but the authors did not provide an analysis or comparison of them.


Suggestions: Consider open-sourcing your code.

**Questions:**

1. The authors emphasize that their work is efficient. Therefore, the experimental section should include comparisons of performance, such as storage and computational overheads at different stages, with existing related works.
2. Is the comparison with related works sufficiently comprehensive? At least, we find that papers ([DCAP: A Secure and Efficient Decentralized Conditional Anonymous Payment System Based on Blockchain] and [AttriChain: Decentralized traceable anonymous identities in privacy-preserving permissioned blockchain]) are related works, but the authors did not provide an analysis or comparison of them.

**Reviewer Confidence:**

3: The reviewer is confident but not certain that the evaluation is correct

**Scope:**

3: The work is somewhat relevant to the Web and to the track, and is of narrow interest to a sub-community

---

### Official Review · Reviewer_G34r · 2024-11-29

**Novelty:** 4
**Technical Quality:** 4

**Review:**

This paper introduces a novel decentralized identity (DID) system aimed at achieving both full anonymity and accountability. The proposed system employs (n,t)-threshold secret sharing to ensure that no single authority can independently obtain a user's real identity.

•	Quality:  The overall quality of the work is solid, with a clear presentation of the system's goals and algorithms. However, the evaluation section has notable shortcomings. First, the experiments are insufficient, particularly in demonstrating the system's performance after deployment on a real blockchain. Second, while several design goals are outlined, only efficiency metrics for the system's algorithms are evaluated, leaving other goals unaddressed.

•	Clarity:  The paper effectively discusses the limitations of existing DID solutions. However, the first figure is overly complex and challenging to interpret. The security proof section is relatively brief and lacks depth, and there is no clear explanation supporting the claim that the system's complexity is O(1).

•	Originality:  The work gives the impression of combining existing technologies—such as distributed anonymous credentials (DACs) and anonymous signatures—without demonstrating substantial novelty. The authors need to explicitly explain what sets this system apart and why it outperforms existing approaches.

•	Significance:  While the work has potential relevance in the blockchain domain, its significance is unclear without stronger experimental validation and clearer articulation of its advantages over existing solutions.

Pros

1.	Provides a thorough comparison of existing solutions, highlighting why they fail to achieve the stated design goals.

2.	Offers detailed explanations for each phase of the system's workflow.

Cons

1.	The evaluation section is underdeveloped. It lacks comparative experiments with previous solutions and does not sufficiently address how the system achieves full anonymity, traceability, and unforgeability.

2.	The novelty of the system is not well-articulated. The paper does not adequately explain why secret sharing is a critical component and how it distinguishes this work from existing approaches.

**Questions:**

1.	Could you elaborate on the novelty of your work? What specific contributions does your system offer those existing solutions lack? Currently, it appears to combine established techniques without introducing significant innovations.

2.	Are there any comparative experiments between FADID-TT and other systems listed in Table 1? If not, could you provide these results or explain why they are missing?

3.	Could you provide additional experimental evidence demonstrating how FADID-TT achieves full anonymity and traceability?

4.	The paper claims that the system achieves O(1) complexity, but this is not substantiated in the text. Could you provide a detailed explanation or proof of this claim?

**Reviewer Confidence:**

1: The reviewer's evaluation is an educated guess

**Scope:**

4: The work is relevant to the Web and to the track, and is of broad interest to the community

---

### Official Review · Reviewer_eRxc · 2024-11-30

**Novelty:** 4
**Technical Quality:** 3

**Review:**

Pros: This paper presents FADID-TT, a fully anonymous decentralised identity system with threshold traceability implemented on a practical blockchain platform. This work demonstrates the rigor of the technology through comprehensive system design, security proofs and experimental validation.
Cons: There is limited discussion of practical deployment challenges, particularly with respect to integration with existing DID systems. The computational overhead of cryptographic operations may limit scalability.

**Questions:**

1.	How does the system perform in terms of computational efficiency and latency across tens of thousands of users?
2.	What are the potential challenges of integrating FADID-TT with existing DID standards or blockchain infrastructure?
3.	How can the system ensure that ADID committees cannot collude to abuse tracking capabilities without compromising the privacy of honest users?

**Reviewer Confidence:**

3: The reviewer is confident but not certain that the evaluation is correct

**Scope:**

3: The work is somewhat relevant to the Web and to the track, and is of narrow interest to a sub-community

---

### Official Review · Reviewer_dJC2 · 2024-12-01

**Novelty:** 6
**Technical Quality:** 6

**Review:**

The paper proposes FADID-TT, a decentralized identity system supporting full
anonymity and threshold traceability using blockchain. It integrates secret
sharing, anonymous signatures, and decentralized anonymous credentials to
protect privacy while ensuring accountability. FADID-TT eliminates public key
management overhead, resists link analysis, and supports public verifiable
tracing via zero-knowledge proofs.

**Pros**

- Introduced a new concept of full anonymity to balance privacy and
  accountability. It's an innovative combination of privacy and accountability
  mechanisms.
- The motivation is well justified with a clear problem statement and
  objectives.
- The proposed system is evaluated through a comprehensive security analysis and
  performance evaluation.

**Cons**

- Potential vulnerability to collusion among committee members, reducing
  practicality in real-world scenarios.
- The roles of ADID committee members are unclear; concrete examples (e.g.,
  government, banks) would clarify their real-world applicability.
- The registration phase does not explicitly address verification of legitimate
  identities, raising concerns about potential impersonation or fake identities.
- Limited focus on scalability with increasing users.

**Questions:**

1. Could you provide concrete examples of the roles within the ADID committee?
   For instance, who would the committee members be in practice—government
   agencies, financial institutions, etc.?
2. In the registration phase, how does the ADID committee ensure that the
   identity $s$ is legitimate and truly bound to the requester's real identity?
   What measures prevent malicious users from registering with fake identities
   or impersonating others?
3. Why does the runtime of "UKeyVerify" scale linearly with the threshold $t$,
   while "UKeyRecon" remains constant in terms of both the threshold $t$ and the
   number of committee nodes $n$? Could you clarify the reasoning behind this
   difference?

**Update:**

Thank you for your response.

For **Q2**: In the "Compliance Checks" procedure, does the committee require plain-text identifiers, such as SSNs, for verification? This seems to contradict the paper (lines 568-572).

For **Q3**: May I request consistent legends between Figures 4 and 5? The current legends seem misleading.

**Reviewer Confidence:**

2: The reviewer is willing to defend the evaluation, but it is likely that the reviewer did not understand parts of the paper

**Scope:**

4: The work is relevant to the Web and to the track, and is of broad interest to the community

---

### Official Review · Reviewer_kxKn · 2024-12-02

**Novelty:** 6
**Technical Quality:** 6

**Review:**

### Abstract
This paper introduces FADID-TT, a fully anonymous decentralized identity (DID) system that balances privacy and accountability through threshold tracing. It uses techniques like secret sharing, anonymous signatures, and decentralized anonymous credentials (DACs) to ensure anonymity, unlinkability, and resistance to link attacks, while simplifying public key management. FADID-TT also supports threshold-based tracing for accountability, with privacy and verifiability built in. The system is tested on Hyperledger Fabric and Ethereum, showing efficient performance with low overhead.

### Reasons to accept
- Successfully integrates full anonymity with threshold traceability in a decentralized identity (DID) system.
- A significant contribution to scalability by reducing user public key management complexity from O(n) to O(1).

### Reasons to reject
- Not answering a specific Web-related scientific research challenge.
- The scheme requires a set of non-colliding registration entities (i.e., ADID committee) that work collaboratively together.

### Comments for authors
- The paper looks like a fully matured paper which is written in a very good shape and structure. It addresses important challenges in PETs. However though the authors mentioned Web3 as one of the main areas to deploy their scheme, this author is not convinced about the direct relation between the paper and the WWW venue.
- Regarding choosing members of the ADID committee, the scheme assumes that the committee operates under an "honest-but-curious" model. If this assumption fails (e.g., collusion between a majority of committee members), the system’s privacy and security properties could be compromised. This might result in game theory problems leading to strategic misbehavior during tracing or even susceptibility to Sybil attacks.

**Questions:**

- Could you elaborate more on how the scheme presented is directly solving a Web-specific challenge?
- How committee members are selected to prevent possible colluding?

**Reviewer Confidence:**

2: The reviewer is willing to defend the evaluation, but it is likely that the reviewer did not understand parts of the paper

**Scope:**

1: The work is irrelevant to the Web